# Regulation of Satiety by *Bdnf-e2*-Expressing Neurons through TrkB Activation in Ventromedial Hypothalamus

**DOI:** 10.3390/biom13050822

**Published:** 2023-05-11

**Authors:** Pengcheng Chu, Wei Guo, He You, Bai Lu

**Affiliations:** 1School of Pharmaceutical Sciences, IDG/McGovern Institute for Brain Research, Tsinghua University, Beijing 100084, China; 2School of Life Sciences, Tsinghua University, Beijing 100084, China; 3Stellenbosch Institute for Advanced Study (STIAS), Wallenberg Centre, 10 Marais Street, Stellenbosch 7600, South Africa

**Keywords:** brain-derived neurotrophic factor, *Bdnf* promoters, ventromedial hypothalamus, obesity, hyperphagia

## Abstract

The transcripts for *Bdnf* (*brain-derived neurotrophic factor*), driven by different promoters, are expressed in different brain regions to control different body functions. Specific promoter(s) that regulates energy balance remain unclear. We show that disruption of *Bdnf* promoters I and II but not IV and VI in mice (*Bdnf-e1^−/−^*, *Bdnf-e2^−/−^*) results in obesity. Whereas *Bdnf-e1^−/−^* exhibited impaired thermogenesis, *Bdnf-e2^−/−^* showed hyperphagia and reduced satiety before the onset of obesity. The *Bdnf-e2* transcripts were primarily expressed in ventromedial hypothalamus (VMH), a nucleus known to regulate satiety. Re-expressing *Bdnf-e2* transcript in VMH or chemogenetic activation of VMH neurons rescued the hyperphagia and obesity of *Bdnf-e2^−/−^* mice. Deletion of BDNF receptor TrkB in VMH neurons in wildtype mice resulted in hyperphagia and obesity, and infusion of TrkB agonistic antibody into VMH of *Bdnf-e2^−/−^* mice alleviated these phenotypes. Thus, *Bdnf*-*e2-*transcripts in VMH neurons play a key role in regulating energy intake and satiety through TrkB pathway.

## 1. Introduction

Energy homeostasis in animals is achieved by a balanced processes of energy intake and energy expenditure [1,2]. Elevated energy intake or reduced energy expenditure could lead to superfluous energy storage, fat accumulation, and eventually obesity [3]. Featuring a rising prevalence in modern society, obesity brings many clinical problems. Therefore, it is important to study the mechanisms of energy homeostasis, and to figure out the causes and mechanisms of obesity which results from genetic and environmental factors.

Brain-derived neurotrophic factor (BDNF) is a risk gene of obesity, since some single nucleotide polypeptides (SNPs) of this gene are linked with obesity [4]. Given that mutations in *Bdnf* gene and BDNF haploinsufficiency result in obesity in both humans and rodents, the causal relationship between BDNF and energy homeostasis was widely accepted [5,6,7,8,9]. However, the mechanisms by which BDNF regulates energy intake and energy expenditure [10], and the neural circuits underlying such regulation, remain poorly understood.

In addition to regulating energy balance, BDNF is also involved in a plethora of functions in different parts of the brain [11,12,13]. This may be achieved by the unique genomic structure: the same BDNF protein could be translated from nine *Bdnf* transcripts originating from nine promoters. In theory, this unique structure allows different promoters to drive BDNF expression with different spatial and temporal patterns. Indeed, this hypothesis has been at least partially validated through mutant mouse lines with disruptions in specific *Bdnf* promoters (see [14] for review). For examples, deficiency in promoter IV-driven BDNF expression led to cognitive dysfunctions, whereas that of promoter I and promoter II resulted in aggressive behaviors [15,16,17].

Since BDNF takes part in both energy intake and energy expenditure [10], it is conceivable that these bidirectional functions could be regulated separately by different promoters. Consistent with this idea, we previously reported that promoter I expressed in lateral hypothalamus (LH) was involved in the regulation of brown adipose tissues (BAT)-mediated thermogenesis which is a part of energy expenditure [18]. However, promoters that specifically regulate energy intake remain to be established. Although some brain nuclei express BDNF to regulate energy intake [10,19], it is not known which promoters are specifically used in these nuclei to regulate energy intake and what are the molecular and neuronal mechanisms for these promoters to function. 

In the present study, we attempted to delineate specific promoter(s) involved in energy intake and/or energy expenditure, using four *Bdnf* promoter mutant lines. Body weights and food intake were measured under different housing conditions and diets for these mice. BAT-mediated thermogenesis was examined by measuring rectal temperatures while exposing the animals to cold. By examining enhanced green fluorescent protein (eGFP) linked to the expression of *Bdnf-e2*, we identified the hypothalamic nuclei critical for *Bdnf-e2* regulation of hyperphagia. We found that re-expressing *Bdnf-e2* transcript or injection of a TrkB agonistic antibody in a hypothalamic nucleus could reverse the hyperphagia and obesity phenotypes in *Bdnf-e2^−/−^* mice to the levels of wild-type (WT) mice. Chemogenetic activation of the hypothalamic neurons could also reduce the energy intake and body weight of *Bdnf-e2^−/−^* mice. Taken together, our results provide important new insights into mechanisms and brain circuits by which different *Bdnf* promoters regulate energy intake/expenditure.

## 2. Materials and Methods

### 2.1. Animals

*Bdnf^+/−^*, *Bdnf-e1^−/−^*. *Bdnf-e2^−/−^*, *Bdnf-e2^−/−^* and *Bdnf-e6^−/−^* were generated and genotyped as previously described [18]. Unless otherwise stated, all mice are housed in groups at controlled temperature (22 °C) with a 12:12 h light-dark cycle and with *ad libitum* access to water and a standard chow diet (SCD, Jiangsu Xietong). *Bdnf-*KO and TrkB-flox mice were bred in Animal center of Tsinghua University. *Bdnf-ires-Cre* mice were from Dr. Wei L. Shen (ShanghaiTech University) as a gift. Ob/Ob mice were bought commercially (Beijing HFK). Diet-induced obese mice were developed by feeding high-fat diets (60% fat, Research Diets, New Brunswick, NJ, USA, #D12492) to commercial C57/B6J mice for more than one month. All animal experiments were performed in accordance with the Institutional Animal Care and Usage Committee of Tsinghua University (AP# 16-LB2).

### 2.2. Food Consumption Test

After 1-week habituation in experimental room, group-housed mice were separated into single cages for 12-day food consumption test. Food pellets were prepared in advance, added to hoppers in the beginning, and weighted daily with an electronic balance. Generally, the food pellet was gnawed to a small sphere (about 0.2 g for standard chow diet), then fell off from the hopper. For the consumption test of high-fat diet (HFD), SCD was provided in the first 6 days and HFD with 45% fat (Research Diets, #D12451) in the last 6 days.

### 2.3. Body Composition Analysis

A magnetic resonance imaging technique (EchoMRI) was used to measure the body composition of live mice, in which fat and mean mass were analyzed. 

### 2.4. Cold Tolerance Test

Group-housed mice were separated into transparent plastic boxes for habituation of 30 min, and then rectal temperature was measured (BIOSEB). After that, mice within these boxes were put into a 4 °C incubator, and after 1 h, rectal temperature was measured again. The used boxes with air holes were 12 cm long, 8 cm wide, and 5 cm high.

### 2.5. Western Blot

Tissues or cells were harvested and homogenized in lysis buffer (150 mM NaCl, 2 mM EDTA, 20 mM HEPES (pH 7.4), 1% NP-40, 1% Deoxycholic acid, 0.1% SDS, and protease and phosphatase inhibitor cocktails). Protein extracts were electrophoresed on SDS-PAGE gels and transferred to PVDF membranes. Transferred membranes were blocked in 5% BSA TBST buffer for 1 h, and then incubated with primary antibodies (1:1000) at 4 °C overnight. Secondary antibodies (HRP, 1:5000) were incubated with membranes at room temperature for 1 h. Pierce ECL Western Blotting Substrate (Thermo Scientific, Waltham, MA, USA) was used to visualize bands, which were detected by Tanon 5200 system (Tanon, Shanghai, China) and analyzed by ImageJ. The following primary antibodies were used: anti-TH (rabbit, 1:500, CST, #2719), anti-ERK (rabbit, 1:1000, CST, #4695), anti-pERK (rabbit, 1:1000, CST, #4730), anti-GAPDH (mouse, 1:1000, YESASEN, #30201), anti-β-Actin (mouse, 1:1000, YESASEN, #30101) and anti-β-tubulin (rabbit, 1:1000, YESASEN, #30301).

### 2.6. Immunohistochemistry

After deep anaesthetization with avertin, mice were transcardially perfused with PBS, followed by 4% PFA. Brains were postfixed in 4% PFA overnight, cryoprotected in 30% sucrose PBS, and cut coronally in 40 μm serial sections with a cryostat. Brain sections that were collected in 24-well plates, were incubated with blocking buffer for 1h at room temperature, and then incubated with primary antibodies at 4 °C overnight. The following primary antibodies were used: anti-GFP (chicken, 1:1000, AVES, #GFP-1010), anti-NeuN (mouse, 1:1000, Millipore, Burlington, MA, USA, #MAB377), and anti-TrkB (rabbit, 1:50, Santa Cruz, #377218). After three washes in PBS, sections were incubated with fluorescent secondary antibodies containing 5 μg/mL Hoechst for 1 h at room temperature. Sections were washed three times by PBS, mounted onto slides in a mounting medium, and coverslipped for imaging with a confocal microscope (Zeiss LSM780). 

### 2.7. Real-Time qRT-PCR

RNA extraction and cDNA reverse transcription were all performed under the requirements of the extraction kit (Vazyme, Nanjing, China, #RC101, #P611). qPCR was performed with Bio-Rad CFX Connect, according to the protocol of ChamQ SYBR qPCR Master Mix (Vazyme, #Q711). qPCR primers were listed in Appendix A, and Actb was the reference gene.

### 2.8. Construction and Packaging of e2-BDNF-Myc

The DNA sequence for *e2*-BDNF transcript (about 4.3 kb) was constructed by homologous recombination (Hieff Clone^®^ Plus Multi One Step Cloning Kit, YEASEN, Shanghai, China #10912ES10) from exon 2 and exon 10 at the alternative splicing site, and then the sequence of Myc tag was inserted into the 3′ terminus of coding region. pAAV-miniCMV-EGFP (PackGene, Guangzhou, China, #K104) was used as the backbone to generate the AAV plasmid of *e2*-BDNF-Myc. The plasmid was transfected into HEK293T cells, and 48 h later, cell lysates were harvested to detect the protein expression using an anti-Myc antibody in western blot experiment. AAV2-miniCMV-*e2*-BDNF-Myc was packaged by PackGene.

### 2.9. Peripheral Treatment of TrkB Agonist Antibody 

TrkB agonist antibody or control IgG was dissolved in PBS to the working concentration of 0.5 mg/mL (unless mentioned elsewhere). These drugs were infused into the tail vein of mice with a fixator, and the final doses were labeled in the main text. For the test of multiple injections, the dose of 1 mg/kg was used, and mice received four intravenous injections, with the frequency of one injection a week.

### 2.10. Stereotactic Injection of Adeno-Associated Virus and TrkB Agonistic Antibody

The brain stereotaxic apparatus was assembled with a 5-uL Hamilton syringe plus a 33-gauge needle that inhaled the appropriate amount of adeno-associated virus (AAV) in the dilution of 1 × E12 titer. After anesthetization with Avertin (200 mg/kg), *Bdnf-e2^−/−^* mice were settled in apparatus, and then received bilaterally 320ul administration of AAV-*e2*-BDNF-Myc or AAV-GFP into VMH (AP −1.60 mm, DV −5.70 mm, ML ±0.30 mm). For DVC rescue experiments, 300 μL AAV-*e2*-BDNF-Myc or AAV-GFP was injected into DVC (AP −7.60 mm, DV −1.80 mm, ML 0.00 mm). AAV-syn-DIO-HM3D(Gq)-mCherry (AAV2/9, OBiO) and AAV-syn-HM3D(Gq)-mCherry (AAV2/9, OBiO) were used to infect *Bdnf-*ires-Cre and *Bdnf-e2*^−/−^ mice respectively. AAV-syn-EGFP-T2A-Cre (AAV2/9, OBiO) and AAV-syn-EGFP-P2A-MCS (AAV2/8, OBiO) were used to infect TrkB-flox mice. For antibody injection, 200 nl TrkB-agomab (3.4 mg/mL) or mouse normal IgG (3.4 mg/mL) was administered bilaterally into the VMH of *Bdnf-e2*^−/−^ mice. After the surgery, mice were administrated with metacam (1 mg/kg) for analgesia and put back to their home cages for recovery, till the following experiments.

### 2.11. Data Processing

GraphPad software was used for data processing, in which repeated two-way ANOVA and unpaired Student’s *t*-test were used for statistical analysis. Image data were processed by Image J and ZEN software. Data plotted in all figures were shown in the way of mean ± SEM. ns represents the *p* value of *t*-test is greater than 0.05; * represents the *p* value of *t*-test is less than 0.05; ** represents the *p* value of *t*-test is less than 0.01; *** represents the *p* value of *t*-test is less than 0.001. The statistical results of repeated two-way ANOVA are described in the main text. 

## 3. Results

### 3.1. Body Weight and Food Intake in Promoter-Specific Bdnf Mutant Lines

Using the promoter-specific mutant lines (*Bdnf-e1^−/−^*, *-e2^−/−^*, *-e4^−/−^* and *-e6^−/−^* for disruption of *Bdnf* promoters I, II, IV, and VI, respectively), we attempted to elucidate the contribution of each of these promoters to the regulation of energy homeostasis [15]. In these mice, the promotor-specific transcripts (*Bdnf-e1*, *-e2*, *-e4* and *-e6* transcripts) direct translation of GFP instead of BDNF protein (e.g., *Bdnf-e2^−/−^*, Figure 1a), due to the insertion of an eGFP-STOP cassette after a specific *Bdnf* exon [10,20]. We examined the body weights of these lines at 5 months of age, since the BDNF heterozygous (*Bdnf^+/−^*) mice begin to exhibit obesity at this age (Figure 1b). Interestingly, *Bdnf-e1^−/−^* and *Bdnf-e2^−/−^*, but not *Bdnf-e4^−/−^* and *Bdnf-e6^−/−^*, mice were significantly heavier than WT littermates (Figure 1b). These results suggest that *Bdnf* promoter I and II take part in energy regulation. 

The increase in body weight could be due to an increase in food intake, and/or a decrease in energy expenditure [10]. In group-housed conditions, we previously reported [18] that *Bdnf-e1^−/−^* mice showed no hyperphagia but exhibited deficits of BAT-mediated thermogenesis, a major process of energy expenditure, as early as the onset period of obesity (also see Figure 1e). However, food intake of *Bdnf-e1^−/−^* mice, when tested in single-housed conditions for more than 1 months, was also increased (Appendix A), as reported previously [21]. A concern over single-housed conditions is that social isolation over such a long period of time, which is known to result in social and cognitive deficits [22,23,24], could also lead to metabolic dysfunctions [25,26,27]. Thus, we developed a 12-day method to measure food intake in mice: group-housed mice at the same age were transferred to the experimental rooms for a 7-day habituation in group, followed by individual housing for 12 days while food consumption of standard chow diet (SCD) was measured daily (Figure 1c). This method has reproduced the hyperphagia phenotype of the conventional *Bdnf* heterozygous mutant mice (*Bdnf^+/−^*) mice, as their daily and cumulative food consumption was significantly higher than that of WT littermates during the test, demonstrating that our method is able to detect abnormal energy intake in mutant mice (Figure 1d). 

### 3.2. Bdnf-e1^−/−^ Mice Exhibited Hyperphagia in Isolation but Normal Food-Intake When Reared in Groups

Using the above method, we found that unlike the *Bdnf^+/−^* mice, *Bdnf-e1^−/−^* mice consumed as much food as WT littermates, when reared in groups with SCD (Figure 1e). To ensure the reproducibility of this result, we did two more repeat experiments for *Bdnf-e1^−/−^* mice raised in groups, and one of them was performed by an independent third party in a completely blind fashion (Appendix A). In both repeat experiments, no hyperphagia phenomenon was observed in *Bdnf-e1^−/−^* mice, suggesting normal food intake in these mice under group housing. Next, we examine whether *Bdnf-e1^−/−^* mice showed hyperphagia during the onset period of obesity, which may contribute to development of their obesity. Using SCD and the same method as described above, we measured body weight and food intake of group−housed *Bdnf-e1^−/−^* mice at 10-week-old age. Throughout the test, although *Bdnf-e1^−/−^* mice gradually became heavier than WT littermates (Appendix A), they consumed the same amount of food as WT littermates (Appendix A). This result further suggests that promoter I is not a key promoter in regulating energy intake. 

It is clear that the *Bdnf-e1^−/−^* mice exhibit normal food intake but hyperphagia after long-term social isolation. We next examined whether hyperphagia could also happen when these mice face other environmental changes, such as high-fat diet (HFD). Indeed, when the SCD was switched to HFD (45% fat), the *Bdnf-e1^−/−^* mice ate more than WT littermates (Appendix A). For comparison, no increase in energy intake was found in *Bdnf-e4^−/−^* mice on a HFD or long-term social isolation (Appendix A). This is interesting, because both promoter IV and I regulate BDNF expression in an activity-dependent manner. Thus, in normal conditions *Bdnf-e1^−/−^* mice are obese due to deficits in thermogenesis, although under different environmental challenges such as social isolation or HFD, they may also eat more than WT mice.

### 3.3. Bdnf-e2^−/−^ Mice Are Obese Due to Hyperphagia

As mentioned above, obesity was also observed in *Bdnf-e2^−/−^* mice. With the 12-day method, we found that the daily and cumulative food consumption of *Bdnf-e2^−/−^* mice was both significantly higher than that of WT littermates (Figure 1f), indicating that *Bdnf-e2* deficiency leads to increased food intake. In comparison, *Bdnf-e1^−/−^* and *Bdnf-e4^−/−^* mice consumed the same food as that in WT littermates (Figure 1e and Appendix A). Similar hyperphagia phenotypes in *Bdnf^+/−^* and *Bdnf-e2^−/−^* mice indicate that promoter II is a specific *Bdnf* promoter enabling BDNF to regulate energy intake. 

Although we observed the hyperphagia of 5-month-old *Bdnf-e2^−/−^* mice, it remained to be established whether increased energy intake causes their obesity. Therefore, we examined the changes in energy intake and body weight of *Bdnf-e2^−/−^* mice during the onset phase of obesity (Figure 2a,b). Food intake and body weight were measured daily in mice at the age of around 7 weeks. In the first 6 days, there was no significant difference in either energy intake or body weight between *Bdnf-e2^−/−^* mice and WT littermates. On the 7th day, *Bdnf-e2^−/−^* mice started to consume far more food and thus became significantly heavier than WT littermates. In the following days, both food consumption and body weight of *Bdnf-e2^−/−^* mice were far greater than those of WT littermates. Thus, excessive energy intake may be the major cause of the overweight in *Bdnf-e2^−/−^* mice.

In addition to increased energy intake, obesity may result from a reduced energy expenditure [28]. Our previous study has shown that the obesity of the *Bdnf-e1^−/−^* mice was due primarily to an impaired thermogenesis in BAT (Brown adipose tissue) [18]. Thus, we examined the thermogenic ability of BAT in *Bdnf-e2^−/−^* mice when they started to become overweight at the age of 6~7 weeks. UCP1 (uncoupling protein 1) is a key thermogenic gene in BAT, and its expression level reflects the thermogenic ability of BAT [18]. Western blot experiments showed that the expression level of UCP1 in *Bdnf-e2^−/−^* mice was comparable to that in WT mice, suggesting that disruption of promoter II-driven BDNF expression does not impair BAT-mediated thermogenesis (Figure 2c). In addition, cold stress can stimulate BAT to produce more heat to maintain body temperature, which could be used to directly assess the thermogenic ability of BAT. We measured the rectal temperature of the mice at room temperature as well as that after 1hour exposure in a 4 °C incubator. No significant difference was found between *Bdnf-e2^−/−^* and WT mice in either condition (Figure 2d). These results suggest that BAT-mediated thermogenesis makes a negligible contribution to the development of obesity in *Bdnf-e2^−/−^* mice.

Taken together, the obesity of *Bdnf-e2^−/−^* mice is due to hyperphagia, whereas impairment of BAT-mediated thermogenesis may be a major contributor to the obesity of *Bdnf-e1^−/−^* mice.

### 3.4. Distribution of Bdnf-e2 Transcript in Brain

To determine the brain regions responsible for *Bdnf-e2* regulation of energy intake, we examined the distribution of *Bdnf-e2* transcript in the brain. Since the mutant transcript of promoter II was translated into eGFP (pII-driven eGFP) in *Bdnf-e2^−/−^* mice, the distribution of *Bdnf-e2* transcript could be detected through immunostaining eGFP in *Bdnf-e2^−/−^* brains. 

Given that hypothalamic region expresses BDNF [29] and is known as the center for regulating energy intake [30], we first examined the expression of pII-driven eGFP in hypothalamic nuclei. Interestingly, pII-driven eGFP was highly expressed in the ventromedial hypothalamus (VMH), including the ventrolateral and dorsomedial parts (VMHvl and VMHdm, Figure 3a–f and Appendix A). Meanwhile, some expression of pII-driven eGFP was detected in the paraventricular hypothalamus (PVH), especially in the medial part (PVHm). In contrast, we did not observe detectable immunostaining signals for pII-driven eGFP in the arcuate nucleus (ARC) and dorsomedial hypothalamus (DMH), the nuclei often implicated in food intake [10,31]. Unlike that in *Bdnf-e1* mice, eGFP was not found in lateral hypothalamus (LH) of *Bdnf-e2* mice (Figure 3d and Appendix A). 

In addition to hypothalamic nuclei, the dorsal vagal complex (DVC, including area potrema, nucleus tractus solitarii, and dorsal motor nucleus of the vagus nerve) is regarded as another candidate region for BDNF regulation of energy intake [32]. Therefore, we examined the expression of pII-driven eGFP in DVC and observed some expression there, especially in area postrema (AP) and nucleus tractus solitarii (NTS) (Figure 3h). Furthermore, we counted the cells expressing pII-driven eGFP in VMH, PVH, and DVC. VMH showed the greatest number of positive cells, followed by DVC and then PVH. The high expression of promoter II in VMH and DVC suggests that these two nuclei may be responsible for the regulation of energy intake by *Bdnf*-*e2*. 

### 3.5. Restoration of Energy Intake through Re-Expressing Bdnf-e2 Transcript in VMH but Not DVC in Bdnf-e2^−/−^ Mice

To determine the region where promoter II plays a role in the regulation of energy intake, we designed the rescue experiments that specifically re-expressed *Bdnf*-*e2* transcript in VMH or DVC. Firstly, we constructed an AAV plasmid encoding *Bdnf-e2* transcript with a Myc tag inserted to the C terminus of its coding sequence (Figure 4a and Appendix A), named AAV-*e2*-BDNF-Myc. Then, we verified its expression of BDNF in transfected HEK293 cells with western blot using anti-Myc antibodies (Figure 4a). 

If loss of *Bdnf-e2* transcript in VMH results in hyperphagia of *Bdnf-e2^−/−^* mice, then local re-expression of *Bdnf-e2* transcript in VMH should be able to attenuate this phenotype and obesity. After injection of AAV viruses into VMH (Figure 4b), *Bdnf-e2^−/−^* mice injected with AAV-*e2*-BDNF-Myc, but not control AAV-GFP, started to lose weight as early as the first week of the injection (Figure 4c). Moreover, the body weight of *Bdnf-e2^−/−^* mice receiving AAV-*e2*-BDNF-Myc dropped to the level of WT mice after 4 weeks of infection (Figure 4d). The weight loss included both fat mass and lean mass, which also became the same as those in WT mice after 4 weeks of injection (Figure 4e). Two weeks after the injection of AAV, we also measured the energy intake of these mice for 12 days using the method we developed, and found substantial cut in energy intake of *Bdnf-e2^−/−^* mice injected with AAV-*e2*-BDNF-Myc. Their energy intake was close to that of WT mice (Figure 4f).

We also injected AAV-*e2*-BDNF-Myc or AAV-GFP into DVC of *Bdnf-e2^−/−^* mice. After 4 weeks of injection, there was no difference in body weight between the two groups. The body weight of the mutant mice was still significantly higher than that of WT littermates. Re-expressing *Bdnf-e2* transcript in DVC failed to reduce the body weight of *Bdnf-e2^−/−^* mice (Appendix A). Besides, in the test of food consumption, *Bdnf-e2^−/−^* mice injected with AAV-*e2*-BDNF-Myc into DVC consumed comparable food to those injected with AAV-GFP. Both groups still showed hyperphagia, when compared to WT littermates (Appendix A). Thus, re-expression of *Bdnf-e2* transcript in DVC cannot exempt *Bdnf-e2^−/−^* mice from hyperphagia. These results suggest that VMH, but not DVC, is where *Bdnf* promoter II functions to regulate energy intake and body weight. 

### 3.6. Rescuing Hyperphagia Deficits by Chemogenetic Activation of VMH Neurons 

Since VMH neurons are important for the regulation of satiety and energy intake [10,33,34,35], we examined whether *Bdnf-e2* was expressed in VMH neurons or not. In *Bdnf-e2^−/−^* mice, cells expressing GFP from promoter II in VMH were positive for the immunostaining of a neuronal marker, NeuN protein (Figure 5a–c). These results suggest that promoter II-driven expression of BDNF in VMH neurons to regulate food intake.

Given than the daily energy intake followed the circadian pattern, we analyzed the food intake of *Bdnf-e2^−/−^* mice in the dark and light periods. As shown in Appendix A, *Bdnf-e2^−/−^* mice ate more food in the dark than that in the light, however, their food intake was significantly higher than WT littermates only in the dark period but not in the light period. Then we separated the 12 h of dark period into three stages (4 h a stage), in which the food consumption of WT mice showed an up-and-down pattern. In the last stage, WT mice exhibited satiety with reduced food intake, but *Bdnf-e2^−/−^* mice did not reduced their food intake. To examine whether satiety of *Bdnf-e2^−/−^* mice is really lower than that of WT mice, we did the fasting and refeeding experiments (Appendix A). After fasting of 24 h, we offered food to *Bdnf-e2^−/−^* mice and their WT littermates and collected the data of their food consumption for 8 h. In the first 4 h, *Bdnf-e2^−/−^* mice consumed same food as WT mic. In the second 4 h, WT mice reduced their food intake because of satiety, but *Bdnf-e2^−/−^* mice ate more food than WT mice. These results suggest lack of satiety in *Bdnf-e2^−/−^* mice, which may be due to lack of *Bdnf-e2* expression derived from promoter II in VMH neurons.

As disruption of promoter II-driven *Bdnf-e2* expression led to the reduction of BDNF protein in VMH (Appendix A) which would reduce the frequency of excitatory postsynaptic currents [36], and application of BDNF enhanced neuronal activity in this nucleus [37], neuronal activity in VMH may be deficient in *Bdnf-e2^−/−^* mice. In the following study, we examined the effects of chemogenetic activation of VMH neurons on the energy intake and body weight of *Bdnf-e2^−/−^* mice (Figure 5d). After 4-week recovery from injection of AAV-syn-HM3D(Gq)-mCherry into VMH, *Bdnf-e2^−/−^* mice received intraperitoneal administration of CNO (1mg/kg) or saline. A single treatment of CNO reduced the food intake of above-mentioned *Bdnf-e2^−/−^* mice in the following 4 and 24 h (Figure 5e). Body weight of *Bdnf-e2^−/−^* mice with AAV infection was also significantly reduced after a day of CNO treatment, when compared with saline group (Figure 5f). For *Bdnf-e2^−/−^* mice without AAV infection, neither body weight nor food intake was reduced by CNO treatment (Appendix A). These results indicate that activating VMH neurons restores energy intake in *Bdnf-e2^−/−^* mice which further reduces their body weight. 

We also determined the effects of activating *Bdnf*-positive neurons on food intake in mice with normal expression of BDNF. AAV-syn-DIO-HM3D-mCherry was injected into VMH of *Bdnf-ires-cre* mice, so that HM3D was expressed on BDNF-positive neurons in VMH (Figure 5g). After chemogenetic activation these neurons with CNO, food intakes of following 4 h and 24 h were both significantly reduced (Figure 5h,i), indicating the importance of neurons expressing BDNF in the regulation of food intake. 

### 3.7. Rescuing Hyperphagia Deficits by Activation of VMH TrkB

Since BDNF is known to regulate energy metabolism through its receptor TrkB [10,31,38,39], we examined the expression of TrkB in *Bdnf*-*e2*-expressing VMH cells. Double immunostaining data showed that TrkB were expressed in *e2*-BDNF positive neurons (Figure 6a–c), raising the possibility that BDNF expressed from promoter II in VMH neurons was secreted to activate TrkB there to regulate energy intake. If so, selective suppression of TrkB in VMH should be able to alter energy intake. Therefore, we deleted TrkB in VMH neurons through injection of AAV-syn-EGFP-T2A-Cre into VMH of TrkB^flox/flox^ mice, to see if it recaptured the phenotypes in *Bdnf-e2^−/−^* mice (Figure 6d–f). After 4 weeks of virus injection, TrkB^ΔVMH^ mice had heavier body weight than mice injected with control virus (AAV-syn-EGFP-P2A-MCS) (Figure 6e). Besides, TrkB^ΔVMH^ mice showed more food consumption than control mice (Figure 6f). These results indicate that local deletion of TrkB gene in VMH increased energy intake in *Bdnf-e2^−/−^* mice.

We then investigated whether hyperphagia and obesity in *Bdnf-e2^−/−^* mice could be ameliorated by locally activating TrkB in the VMH. To accomplish this, we administered a TrkB agonist antibody (TrkB-ago) or control IgG directly into the VMH of *Bdnf-e2^−/−^* mice and monitored changes in body weight and food consumption (Figure 6g–i). *Bdnf-e2^−/−^* mice treated with TrkB-ago showed a significant decrease in body weight compared to those treated with control IgG within the first 4 days (Figure 6h). Furthermore, during this time period, food consumption was significantly lower in *Bdnf-e2^−/−^* mice treated with TrkB-ago compared to those treated with control IgG (Figure 6i). However, from day 4–6, there was no significant difference in daily food consumption between the two groups, and the body weight showed a trend of recovery (Figure 6i). This could be attributed to the diffusion/metabolism of TrkB-ago in the VMH, as TrkB-ago was undetectable after 6 days. Therefore, a single administration of TrkB-ago into the VMH could temporarily alleviate the energy intake and obesity deficits in *Bdnf-e2^−/−^* mice.

Given that VMH is also involved in the regulation of anxiety, one concern is that the reduction of food intake and body weight by TrkB-ago in *Bdnf-e2^−/−^* mice may be due to increased anxiety after TrkB activation. Therefore, we examined the effects of TrkB-ago on anxiety state of *Bdnf-e2^−/−^* mice with open field test 3 days after VMH injection (Appendix A). No significant difference was observed in center time, total activity, mean speed or percent center distance between TrkB-ago and control IgG groups (*Bdnf*-*e2*^−/−^ mice treated with control IgG). These results indicate that injection of TrkB-ago into VMH did not induced anxiety, so that its rescuing effects on energy intake is more likely to be achieved by the satiety regulation of VMH. Taken together, our results suggest that BDNF expressed from promoter II-driven transcription and activation of TrkB are essential for VMH neurons to regulate energy intake.

As our previous study had shown that TrkB-ago is capable of penetrating through blood brain barrier [40], and peripheral administration of TrkB-ago (1 mg/kg) enhanced the phosphorylation of ERK protein, a downstream target of TrkB, in entire hypothalamus (Figure 7a), we test whether peripheral injection of TrkB-ago could also rescue the deficiencies of *Bdnf-e2^−/−^* mice. After a single intravenous injection of TrkB-ago or IgG (1 mg/kg), *Bdnf-e2^−/−^* mice administered with TrkB-ago showed a significant decrease in body weight and food intake, compared with those receiving IgG administration (Figure 7b). Similar effects were also observed in hyperphagic leptin-deficient (*Ob/Ob*) mice (Figure 7c). This finding is consistent with two previous studies showing that leptin can regulate homeostatic feeding through TrkB signaling in a few of hypothalamic regions [31,41]. In addition, Tsao et al. reported that TrkB agonist NT-4 is functional in rescuing obesity phenotype in (DIO) models [42]. Similarly, a single peripheral injection of TrkB-ago induced a weight loss of approximately 18% in diet-induced obese (DIO, 60% fat HFD) mice, and four injections (once a week) almost reduced their body weight to the level of WT mice which could last for at least 3 weeks (Figure 7d). Taken together, these results suggest a potential application of TrkB-ago in the treatment of obesity.

## 4. Discussion

In the present study, we found that promoter II, rather than the other three promoters that are also highly expressed in brains, specifically regulates energy intake rather than thermogenesis through expressing BDNF in neurons of ventromedial hypothalamus. As BDNF expressed from promoter II in VMH regulated energy homeostasis through its receptor TrkB, peripheral injection of a TrkB agonist antibody activated TrkB signaling in hypothalamus and restored the food intake and body weight of *Bdnf-e2^−/−^* mice.

Because long-term social isolation used in traditional methods can cause metabolic and cognitive deficits among others [25,26,27], we attempted to develop a relatively fast and stable method to investigate hyperphagia: mice were raised in group-housed condition till adulthood but food intake was measured in a 12-day single housed condition. Using the 12-day single-housed method, we examined food intake in the 4 *Bdnf* promoter mutant lines which were reared together with WT littermates after weaning. We found that only *Bdnf-e2^−/−^* mice, but not *Bdnf-e1^−/−^* nor *Bdnf-e4^−/−^* and *Bdnf-e6^−/−^* mice exhibited the hyperphagia phenotype during and after the onset period of obesity. These results indicate that promoter II, not other promoters, has a key role in the regulation of energy intake. This is supported by studies of human single nucleotide polypeptide (SNP): some *Bdnf* SNPs and high DNA methylations on this promoter are associated with less expression from promoter II and obesity [43,44,45]. Although promoter II is involved in the regulation of food intake, its roles in other aspects of energy homeostasis remain to be explored. When *Bdnf-e2^−/−^* mice were at the onset period of obesity, we found that these mice did not have thermogenesis deficit and their locomotor activity was comparable to that of WT littermates. Hence, our results support the notion that the major function of promoter II is to regulate food intake. 

Among the four promoters (I, II, IV, VI) that are highly expressed in brains [15], promoter I and promoter II belong to the first spatial cluster, while being regulated by a common intronic enhancer [46]. Mice with mutations in these two promoters developed obesity, but mice with mutations in promoter IV or promoter VI did not, suggesting that the first cluster of BDNF is involved in energy homeostasis. The crucial role of this cluster in energy homeostasis is also found in obese WAGR patients, as the shortest deletion of BDNF gene only includes the first cluster [8]. When housed in standard conditions (in groups and with chow diets), *Bdnf-e1^−/−^* mice exhibited deficits of BAT thermogenesis [18], whereas *Bdnf-e2^−/−^* mice developed hyperphagia. It should be pointed that *Bdnf-e1^−/−^* mice had normal food consumption in standard conditions, although long-term isolation or high-fat diets induced these mice to eat more than WT mice [21]. Nevertheless, our studies indicate that under normal circumstances, promoter I and promoter II are involved in the regulation of BAT thermogenesis and energy intake, respectively. Taken together, these results also help explain why a single protein BDNF could elicit so many different biologic effects-BDNF derived from different promoters is expressed in different cell types and brain regions, regulated by different factors, and elicits distinct functions. 

An interesting finding is that promoter II driven eGFP was highly expressed in VMH and DVC, but re-expression of *Bdnf-e2* transcript in VMH but not DVC rescued hyperphagia and obesity phenotypes of *Bdnf-e2^−/−^* mice. A critical question is: which *Bdnf* transcript(s) in which brain nuclei (VMH, DVC, PVH) is the key for what aspect of energy regulation? The role of VMH BDNF in food intake is known, since selective deletion of *Bdnf* gene in VMH of mice by the Cre-lox system resulted in obesity and hyperphagia [38]. However, while BDNF transcripts derived from promoters I, II, and VI were all expressed in VMH [47], BDNF protein level in VMH was significantly reduced in *Bdnf-e2^−/−^* mice. Further, only disruption of promoter II, but not other *Bdnf* promoters, caused hyperphagia in mice. These results support the hypothesis that for food intake, BDNF regulation through promoter II in VMH is the key. Promoter II-driven BDNF expression is also observed in DVC, and the administration of BDNF protein to DVC did reduce energy intake [32]. Surprisingly, re-expressing *e2* transcript in DVC did not exempt *Bdnf-e2^−/−^* mice from hyperphagia. One might speculate either BDNF protein injected in DVC had diffused nearby to elicit its function, or translation of BDNF through *Bdnf-e2* transcript was somehow inhibited in DVC. Future studies using direct deletion of *Bdnf* gene in DVC should help resolve these possibilities. In the above rescue experiments, we chose *Bdnf*-*e2* transcript (exon 2 plus coding sequence) instead of just BDNF coding sequence for the rescuing experiment for the following reasons: First, all AAV plasmids require a 5′ UTR to express protein [48,49]. Second, the UTRs of BDNF mRNAs could influence their subcellular distribution and ultimately affect energy balance [48,50]. Third, wild-type *Bdnf-e2* transcript was lost in *Bdnf-e2^−/−^* mice, so that it is logical to re-express this transcript in these mice. PVH is another brain region where BDNF regulates energy intake, as selective deletion of BDNF in PVH led to energy imbalance [51]. Data in this study and our previous paper [18] showed that promoter IV and promoter VI, but not promoter I or promoter II, were highly expressed in this region. However, *Bdnf-e4^−/−^* and *Bdnf-e6^−/−^* mice exhibited normal body weight. It remains to be resolved which BDNF promoters (other than I, II, VI, VI) in PVH are crucial to regulate energy intake. 

Promoter II-driven *Bdnf* transcript is expressed in VMH neurons, which are overwhelmingly excitatory and are involved in regulating satiety to food. Without BDNF expression from promoter II in these neurons, *Bdnf-e2^−/−^* mice had food satiety deficiency, resulting in over-eating. Moreover, reduced BDNF led to a reduction in the frequency of excitatory postsynaptic currents in VMH [37], and injection of BDNF into the third ventricle enhanced activity in VMH neurons [38]. Therefore, impairment in satiety of *Bdnf-e2^−/−^* mice maybe resulted from a reduced response of VMH neurons to food consumption. Indeed, chemogenetic activation of VMH neurons in *Bdnf-e2^−/−^* mice inhibited their excess energy intake and reduced their body weight. Activation of BDNF-expressing VMH neurons also reduced food intake in WT mice with normal BDNF expression (Figure 5). Suppression of VMH neurons in WT mice is likely to induce hyperphagia and body weight increase, although such experiments have not been done. Taken together, these results raise the possibility that BDNF, expressed in entire region of VMH, is a suitable marker for neurons which regulate satiety of food. In comparison, SF1, a gene that regulates terminal differentiation of VMH neurons, is regarded as a marker of VMH neurons. In adult mice SF1-positive neurons are distributed only in the dorsomedial part of VMH. However, genetic knockout of *Bdnf* or *Vglut2* in SF1-expressing neurons in VMH did not cause obesity or hyperphagia under normal conditions. Indeed, the expression of BDNF or VGLUT2 in ventrolateral part of VMH seemed undisrupted in these SF1-neuron specific knockout mice [52,53]. Interestingly, the projection of SF1-expressing neurons to paraventricular thalamus (PVT) was reported to be involved in the regulation of satiety [54]. In future studies, *Bdnf-e2* expressing neurons in VMH may be a key to uncover the roles of other projections from or to VMH in the regulation of food intake or satiety.

BDNF receptor TrkB is known to be involved in the regulation of energy intake [19,31,38,39,55]. We found that TrkB was also expressed in VMH neurons expressing promoter II, and deletion of TrkB in the VMH neurons also resulted in hyperphagia and obesity in TrkB conditional knockout mice. Moreover, deficits caused by loss of *e2-*BDNF were rescued by activating TrkB in VMH with TrkB-ago, a TrkB agonistic antibody, which did not induce anxiety when examined in open field test. The reduction of food intake and body weight was not observed 4 days after VMH injection of TrkB-ago when TrkB-ago may be degraded because no immunostaining signals for the antibody was detected 6 days after VMH injection. Besides of VMH, deletion of TrkB in DMH or PVH also led to obesity in mice [31,38,56]. When injected peripherally, TrkB-ago could break through the blood-brain barrier (BBB) and activate downstream signaling in hypothalamus [40], so that peripheral injection of TrkB-ago may be a more powerful way to treat obesity of *Bdnf-e2^−/−^* mice. In this way, we found a drop in both energy intake and body weight in *Bdnf-e2^−/−^* mice after intravenous injection of TrkB-ago. This drug also reduced body weight of mice suffering leptin deficiency and diet-induced obesity, implying its extensive anti-obesity effects. 

## Figures and Tables

**Figure 1 biomolecules-13-00822-f001:**
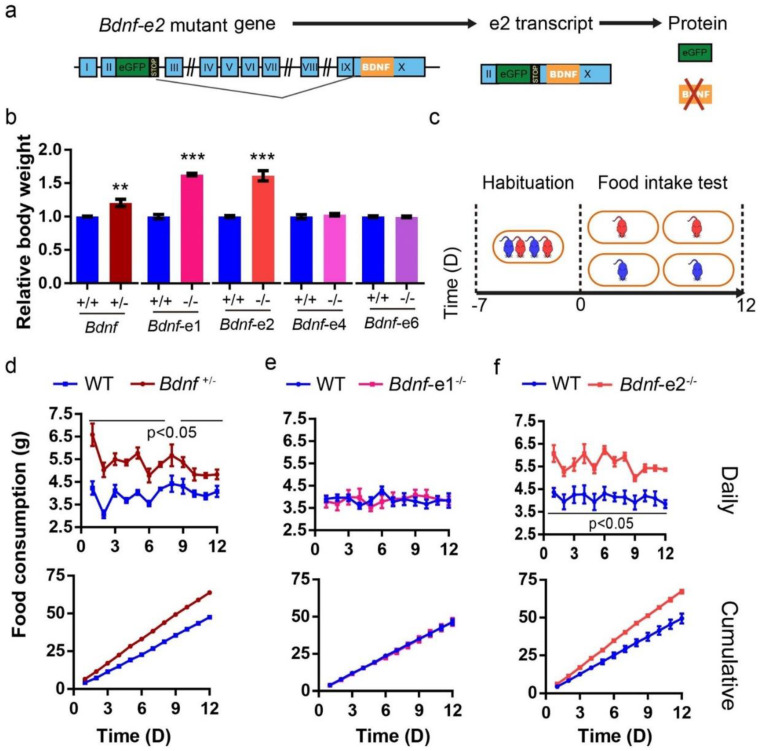
Hyperphagia caused by the disruption of *Bdnf* promoter II. (**a**) Schematic diagrams for the genomic structure of *Bdnf*-*e2*^−/−^ mice. A GFP gene with multiple STOP codons is inserted after *Bdnf* exon II, so that promoter II will drive the expression of GFP instead of BDNF. (**b**) Relative body weight of *Bdnf^+/−^* (n = 6~7 animals), *Bdnf-e1^−/−^* (n = 6~8), *Bdnf-e2^−/−^* (n = 4~6), *Bdnf-e4^−/−^* (n = 4~5), and *Bdnf-e6^−/−^* (n = 4~5) mice at 5 months of age. (**c**) Schematic diagrams for food intake measurement in (**d**–**f**). Mutant mice and their WT littermates were group−housed for 7 days for habituation, and then food intake was measured for 12 days in isolated conditions. (**d**) Daily (**top**) and cumulative (**bottom**) food consumption in *Bdnf*^+/−^ mice (n = 4) and WT littermates (n = 4). *p* values of two-way RM ANOVA for daily and cumulative data were both less than 0.001. (**e**) Daily (**top**) and cumulative (**bottom**) food consumption in *Bdnf*-*e1*^−/−^ (n = 7) mice and WT littermates (n = 6). *p* values of two-way RM ANOVA for daily and cumulative data were 0.2642 and 0.1995, respectively. (**f**) Daily (**top**) and cumulative (**bottom**) food consumption in *Bdnf*-*e2*^−/−^ mice (n = 4) and WT littermates (n = 6). *p* values of two-way RM ANOVA for daily and cumulative data were 0.0025 and 0.0035, respectively. Data were plotted as mean ± SEM. *t*-test: **, *p* < 0.01; ***, *p* < 0.001; ns, no significance. Note that *Bdnf*-KO mice exhibited an increase in food intake, compared with WT. However, only *Bdnf*-*e2*^−/−^ mice but not *Bdnf*-*e1*^−/−^ mice exhibited hyperphagia phenomenon during the 12 days.

**Figure 2 biomolecules-13-00822-f002:**
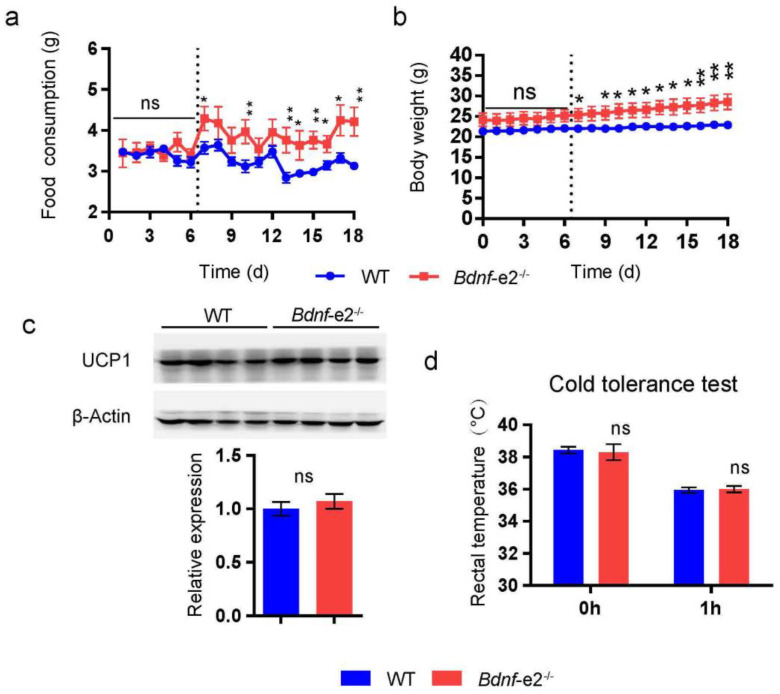
Hyperphagia but normal thermogenesis of *Bdnf*-*e2*^−/−^ mice during the onset period of obesity. Food consumption (**a**) and body weight (**b**) of *Bdnf*-*e2*^−/−^ mice (n = 5) and WT littermates (n = 9) during the onset period of obesity. The dotted vertical line represents the time when the difference in food intake and body weight reached statistical significance with *t*-test. Note: *Bdnf*-*e2*^−/−^ mice exhibited elevated food intake as well as continuous increase in body weight. (**c**) Expression of UCP1 in BAT of *Bdnf*-*e2*^−/−^ mice (n = 4) and WT littermates (n = 4), measured by western blot at the age of 6~7 weeks. ns: no significance by *t*-test. (**d**) Rectal temperature of *Bdnf*-*e2*^−/−^ and WT mice (6~7 weeks) at 0 h and 1 h during cold tolerance test. No thermogenesis deficit was observed in *Bdnf*-*e2*^−/−^ mice. *t*-test: *, *p* < 0.05; **, *p* < 0.01; ns, no significance.

**Figure 3 biomolecules-13-00822-f003:**
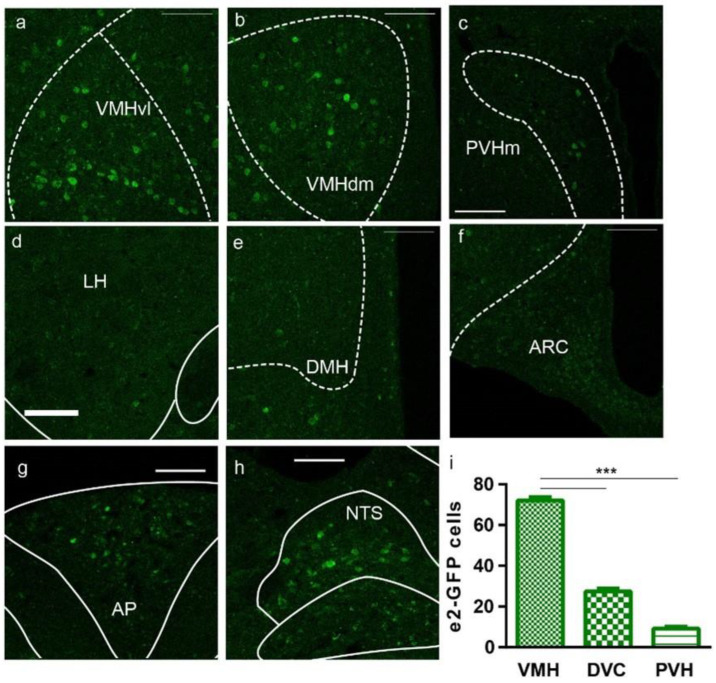
*Bdnf*-*e2* is highly expressed in VMH, less so in DVC and PVH. (**a**–**f**) Selected sections in various regions in hypothalamus. Scale bar: 50 μm. GFP-positive signals represent cells in which *Bdnf*-*e2* expression is disrupted. Note that GFP was highly expressed both in both ventrolateral and dorsomedial parts of ventromedial hypothalamus (VMHvl and VMHdm), less in medial paraventricular hypothalamus (PVHm), but not in arcuate nucleus (ARC), dorsomedial hypothalamus (DMH) or lateral hypothalamus (LH). Dashed lines (white) indicate the separation of nucleus or subnucleus; solid lines (white) indicate neuronal fibers or brain ventricles. (**g**,**h**) Selected sections of area postrema (AP) and nucleus tractus solitarii (NTS) of dorsal vagal complex (DVC). (**i**) Statistics of *e2*-GFP positive cells in PVH, VMH and DVC that were averaged from the coronal sections (n = 4). ***: *p* < 0.001 for *t*-test.

**Figure 4 biomolecules-13-00822-f004:**
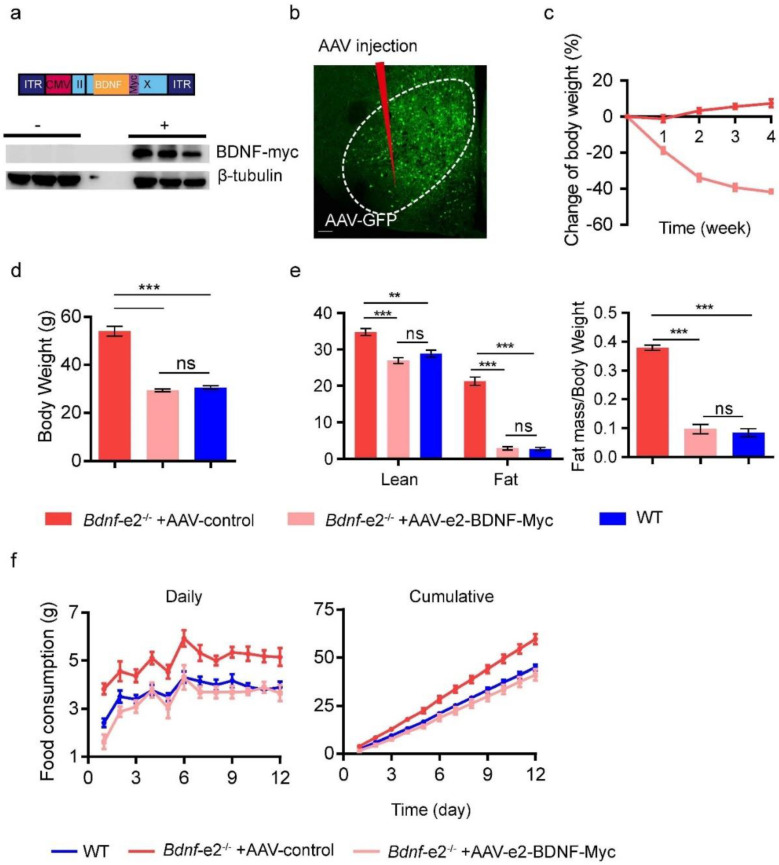
Restoration of body weight and energy intake in *Bdnf*-*e2*^−/−^ mice by re-expressing *e2* transcript in VMH. (**a**) Schematic diagram and western blot of the AAV-*e2*-BDNF-Myc (AAV-*e2*) construct. The expression of *e2*-BDNF-Myc in HEK293T transfected cells was detected with anti-Myc and anti-β-tubulin antibodies. (**b**) AAV injection site and fluorescent signals after injection of control virus (AAV-GFP). Scale bar: 50 μm. Dashed lines (white) indicate the separation of nucleus or subnucleus. (**c**) Change of body weight after injection of AAVs into VMH in *Bdnf*-*e2*^−/−^ mice (n = 5/group). *p* value of two-way RM ANOVA was less than 0.0001. (**d**) Body weight of *Bdnf*-*e2*^−/−^ mice 4 weeks with injection of AAVs into VMH (n = 5/group), compared with WT littermates (n = 6). (**e**) Lean and fat masses and fat contribution to body weight in *Bdnf*-*e2*^−/−^ mice 4-weeks after AAV injection (n = 5/group), compared with WT litters (n = 6). (**f**) Daily and cumulative food consumption in *Bdnf*-*e2*^−/−^ mice receiving injection of *e2*-AAV or AAV-GFP (n = 5/group) into VMH, compared with WT littermates (n = 6). *p* value of two-way RM ANOVA for daily and cumulative data between *e2-*AAV and AAV-GFP groups was 0.0015 and 0.0017, respectively. *t*-test: **, *p* < 0.01; ***, *p* < 0.001; ns, no significance.

**Figure 5 biomolecules-13-00822-f005:**
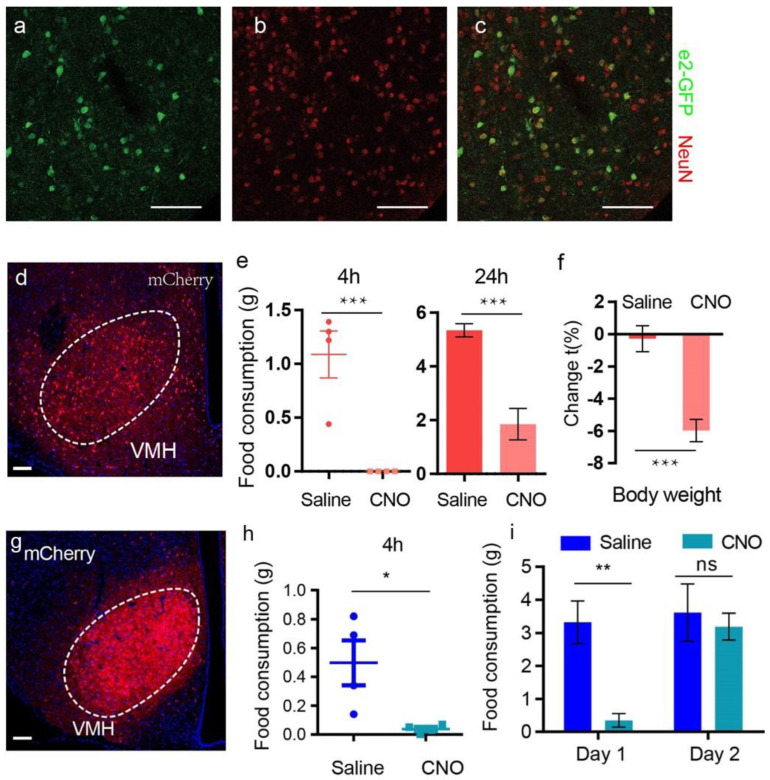
Rescue of less satiation in *Bdnf*-e2^−/−^ mice through chemogenetic activation of VMH neurons. (**a**–**c**) Colocalization of e2-GFP and NeuN in VMH. (**d**) Fluorescent signals of injected AAV-syn-HM3D(Gq)-mCherry into VMH neurons of *Bdnf*-*e2*^−/−^ mice. Dashed lines (white) indicate the separation of nucleus or subnucleus. (**e**) Food consumption (4 h and 24 h) after chemogenetic activation of VMH neurons in *Bdnf*-*e2*^−/−^ mice. (**f**) Changes of body weight after chemogenetic activation of VMH neurons in *Bdnf*-*e2*^−/−^ mice. Scale bar: 50 μm. (**g**) mCherry expression of VMH in *Bdnf*-ires-Cre mice injected with AAV-syn-DIO-HM3D-mCherry. Scale bar: 100 μm. Dashed lines (white) indicate the separation of nucleus or subnucleus. (**h**) 4 h-food consumption after treatment of CNO or saline in mice with HM3D expressed in BDNF positive VMH neurons. n = 4/group. (**i**) Food consumption of the first and second day after treatment of CNO or saline in mice with HM3D expressed in BDNF positive VMH neurons. n = 4/group. *t*-test: *, *p* < 0.05; **, *p* < 0.01; ***, *p* < 0.001; ns, no significance.

**Figure 6 biomolecules-13-00822-f006:**
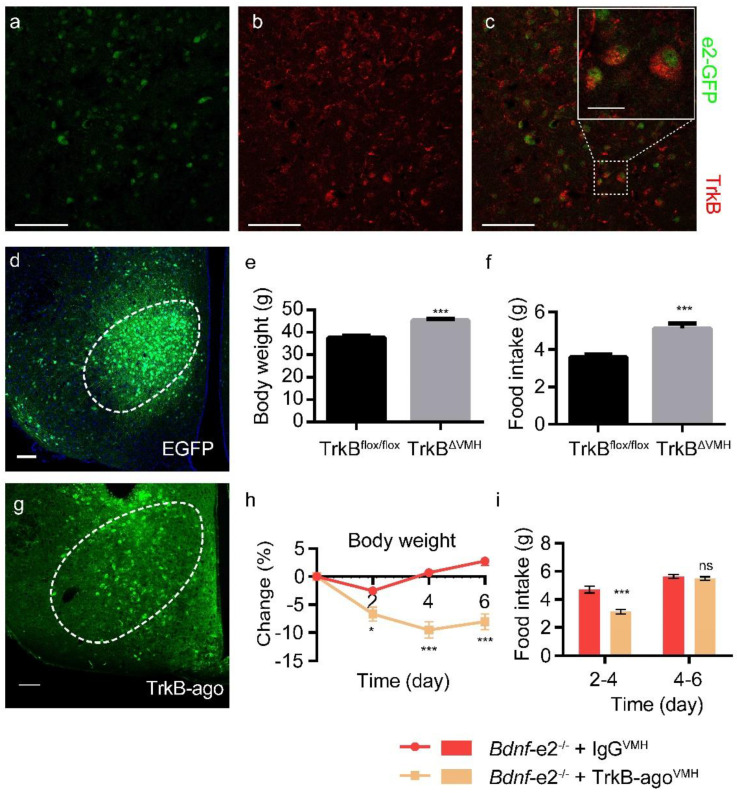
TrkB as a downstream target of BDNF promoter II in VMH to regulate food intake. (**a**–**c**) Colocalization of *e2*-GFP and TrkB in VMH. Scale bar: 50 μm. Upper right panel in (**c**) Enlarged image of dashed rectangle area. Scale bar: 10 μm. (**d**) AAV injection site and fluorescent signals after injection of AAV-Cre into *TrkB^flox/flox^* mice. Scale bar: 50 μm. (**e**,**f**) Energy intake (right) and body weight (left) in *TrkB^flox/flox^* and *TrkB^ΔVMH^* mice after 4 weeks of virus injection. (**g**) Injection site and fluorescent signals representing TrkB-ago injected into VMH of *TrkB^flox/flox^* mice. Scale bar: 50 μm. (**h**) Changes of body weight in *Bdnf*-*e2*^−/−^ mice after IgG or TrkB-ago injection into VMH. (**i**) Daily food intake for Day 2–4 and 4–6 in *Bdnf*-*e2*^−/−^ mice after IgG or TrkB-ago injection into VMH. *t*-test: *, *p* < 0.05; ***, *p* < 0.001; ns, no significance.

**Figure 7 biomolecules-13-00822-f007:**
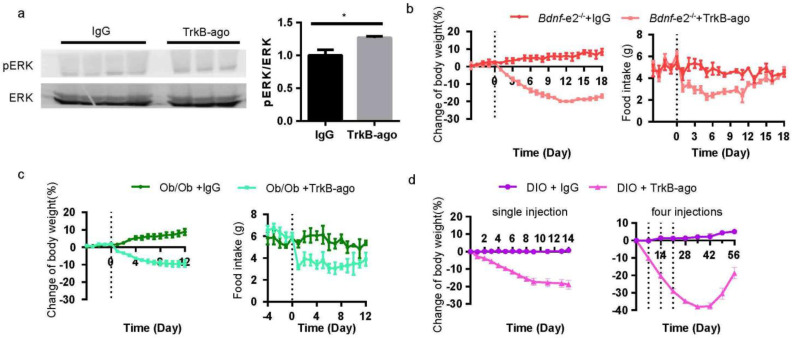
Rescue effects of TrkB agonist antibody in *Bdnf*-*e2*^−/−^ mice and other mouse model with elevated energy intake. (**a**) Western blot of phosphorylated ERK in hypothalamus at 1 day after TrkB-ago intravenous injection (n = 3/group). *t*-test: *, *p* < 0.05. (**b**) Effect of TrkB-ago treatment on body weight (**left**) and food intake (**right**) in *Bdnf−e2^−/−^* mice, with IgG as control (n = 3/group). (**c**) Effect of TrkB-ago treatment on energy intake (**left**) and body weight (**right**) in *Ob/Ob* mice with TrkB-ago or IgG injection (n = 4~5/group). (**d**) Effect of single or four (1 injection/week) TrkB-ago injections on body weight in diet-induced obese mice (n = 3~4/group).

## Data Availability

The data presented in this study are available on request from the corresponding author.

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
