# Peer review of "Regulation of Satiety by Bdnf-e2-Expressing Neurons through TrkB Activation in Ventromedial Hypothalamus"

_biomolecules, 2023, doi:10.3390/biom13050822_

Round 1

Reviewer 1 Report

This manuscript by Chu et al. describes a novel role for the BDNF exon 2 in the regulation of eating behavior and satiety. They show that while exon 1 regulates energy expenditure, exon 2 specifically inhibits satiety, thereby these two exons cooperate regulating different aspects of energy intake. They further show that this effect is mediated by TrkB receptors and that the site of action is the VMH. Numerous complementary up-to-date experimental setups are presented that test different aspects of the hypothesis and their cumulative results provide a convincing demonstration of the action of this BDNF exon. Additional smaller very interesting observations are also presented, such as the role of TrkB in the phenotype of ob/ob mice. Altogether, the results are very interesting and clearly represent novelty, the manuscript is mostly well written (although see below) and the manuscript is clearly worth publishing. I only have a few relatively minor comments that hopefully further improve the paper. 

1.     Fig. 4a: I am not sure what has been the idea of including the exon 2 in front of BDNF in this context. Does the construct also contain the promoter for exon 2 or not? If it does, given that the construct is driven by the strong CMV promoter, it is unlikely that the addition of the exon 2 promoter really makes any difference, the construct is most likely entirely driven by the CMV promoter. If the exon 2 promoter is not included, then it is not clear how this construct differs from a construct where only the coding exon is included. 

2.     Figure 4 and S1 indicate BDNF protein coding region as CDS, which is not spelled out in the legend. By contrast, Fig 1 legend has CDS spelled out, but this abbreviation is not used in the figure, here it is replaced by BDNF. Although a minor thing, please replace CDS with BDNF throughout. 

3.     Figure 6 a-c (by the way, text on lines 337-338 refers to this figure as Extended data Fig. 6a-c, please correct): are these individual confocal sections or projections? It is interesting to note, although consistent with literature, that BDNF and TrkB are quite differentially located in the stained neurons. The authors may want to shortly comment on the colocalization of BDNF and TrkB in at least some cells. Are they always colocalized?

4.     Fig. 6d: Where is EGFP signal coming from? The constructs do not seem to express EGFP.  Please clarify. 

5.     The first report of BDNF expression in the hypothalamic nuclei was in PMID: 7708216, the authors may want to cite this paper. 

6.     Although the text is mostly clearly written, that is not the case for the second-last paragraph of the page 11 (lines 347-358), where several sentences seem to be incomplete and at least very difficult to understand. Also, what is the control that is referred to on line 353? A wild-type mice or what? Please rewrite the entire paragraph. 

7.     There are several references that are listed twice in the reference list. #10=31, 36=44, perhaps others. Please check and correct. 

Altogether, this is a very interesting and well-conducted study that with minor changes should be publishable. 

Author Response

Reviewer #1

This manuscript by Chu et al. describes a novel role for the BDNF exon 2 in the regulation of eating behavior and satiety. They show that while exon 1 regulates energy expenditure, exon 2 specifically inhibits satiety, thereby these two exons cooperate regulating different aspects of energy intake. They further show that this effect is mediated by TrkB receptors and that the site of action is the VMH. Numerous complementary up-to-date experimental setups are presented that test different aspects of the hypothesis and their cumulative results provide a convincing demonstration of the action of this BDNF exon. Additional smaller very interesting observations are also presented, such as the role of TrkB in the phenotype of ob/ob mice. Altogether, the results are very interesting and clearly represent novelty, the manuscript is mostly well written (although see below) and the manuscript is clearly worth publishing. I only have a few relatively minor comments that hopefully further improve the paper.

We thank for the reviewer for the comments that our findings are ‘very interesting and clearly represent novelty’. We have revised the manuscript based on the suggestions by the reviewer and hope the revision could meet his/her satisfaction.

1.Fig. 4a: I am not sure what has been the idea of including the exon 2 in front of BDNF in this context. Does the construct also contain the promoter for exon 2 or not? If it does, given that the construct is driven by the strong CMV promoter, it is unlikely that the addition of the exon 2 promoter really makes any difference, the construct is most likely entirely driven by the CMV promoter. If the exon 2 promoter is not included, then it is not clear how this construct differs from a construct where only the coding exon is included.

Regarding Fig. 4a, there are a number of reasons why we chose the Bdnf e2 transcript (exon 2 plus coding sequence) instead of just the BDNF coding sequence for the rescuing experiment. First, it is important to note that all AAV plasmids require a 5' UTR in addition to coding sequence to express a protein. Second, the untranslated regions (UTRs) of BDNF mRNAs have been shown to dictate their subcellular distribution, which could ultimately affect their functions such as energy balance [1,2]. Last, the Bdnf-e2 transcript was gone in Bdnf-e2-/- mice. Therefore, it is logical to re-express this transcript in these mice for the rescuing experiment. We hope that we have explained our rationale for choosing the Bdnf-e2 transcript for the rescue experiment well.

2.Figure 4 and S1 indicate BDNF protein coding region as CDS, which is not spelled out in the legend. By contrast, Fig 1 legend has CDS spelled out, but this abbreviation is not used in the figure, here it is replaced by BDNF. Although a minor thing, please replace CDS with BDNF throughout.

We have corrected the labeling of BDNF protein coding region in Fig. 4 and S1 and replaced all instances of “CDS” with “BDNF” throughout the manuscript.

3.Figure 6 a-c (by the way, text on lines 337-338 refers to this figure as Extended data Fig. 6a-c, please correct): are these individual confocal sections or projections? It is interesting to note, although consistent with literature, that BDNF and TrkB are quite differentially located in the stained neurons. The authors may want to shortly comment on the colocalization of BDNF and TrkB in at least some cells. Are they always colocalized?

The images presented in Fig. 6 a-c are individual confocal sections, and we have corrected the citation of this figure. We have conducted a new immunostaining experiment and observed that in the region of the VMH, there are neurons expressing both Bdnf-e2 and TrkB as well as neurons expressing TrkB but not Bdnf-e2, indicating that the effects after BDNF secretion may involve both autocrine and paracrine mechanisms in VMH.

4.Fig. 6d: Where is EGFP signal coming from? The constructs do not seem to express EGFP.  Please clarify.

In Fig. 6d, the EGFP signal comes from the AAV-syn-EGFP-T2A-Cre virus used in the experiment to infect TrkBflox/flox mice, leading to the expression of Cre to delete TrkB.

5.The first report of BDNF expression in the hypothalamic nuclei was in PMID: 7708216, the authors may want to cite this paper.

Thank you for your valuable feedback and suggestion to add a reference to PMID: 7708216 in the "Distribution of Bdnf-e2 transcript in brain" section of our manuscript. We have now made the necessary revision, and the reference is included on line 215.

6.Although the text is mostly clearly written, that is not the case for the second-last paragraph of the page 11 (lines 347-358), where several sentences seem to be incomplete and at least very difficult to understand. Also, what is the control that is referred to on line 353? A wild-type mice or what? Please rewrite the entire paragraph.

We appreciate the reviewer’s comment about the second-last paragraph of page 11 (lines 347-358). We have rewritten the paragraph and noted the control mice as Bdnf-e2-/- mice treated with control IgG.

7.There are several references that are listed twice in the reference list. #10=31, 36=44, perhaps others. Please check and correct.

We have removed the duplicated references.

Altogether, this is a very interesting and well-conducted study that with minor changes should be publishable.

References

  1. You, H.; Chu, P.; Guo, W.; Lu, B. A subpopulation of Bdnf-e1–expressing glutamatergic neurons in the lateral hypothalamus critical for thermogenesis control. Molecular Metabolism 2020, 31, 109-123, doi:10.1016/j.molmet.2019.11.013.
  2. You, H.; Lu, B. Diverse Functions of Multiple Bdnf Transcripts Driven by Distinct Bdnf Promoters. 2023, 13, 655.

Reviewer 2 Report

The work from Chu et al., evaluates the role of different BDNF transcripts in the regulation of energy intake, BAT thermogenesis and overall in obesity. The study is well written and employs a combination of mouse genetics complemented with chemogenetics to distinguish the role of specific BDNF transcripts in specific cell types. The study also provides mechanistic insight deleting TrkB in VMH neurons, demonstrating the role of BDNF-TrkB signaling pathway regulating energy intake in this subset of neurons. 

I do not have major nor minor comments for this paper, besides editing the title, which is not totally clear. 

I would accept the manuscript in the current format. 

Author Response

Reviewer #2

The work from Chu et al., evaluates the role of different BDNF transcripts in the regulation of energy intake, BAT thermogenesis and overall in obesity. The study is well written and employs a combination of mouse genetics complemented with chemogenetics to distinguish the role of specific BDNF transcripts in specific cell types. The study also provides mechanistic insight deleting TrkB in VMH neurons, demonstrating the role of BDNF-TrkB signaling pathway regulating energy intake in this subset of neurons.

I do not have major nor minor comments for this paper, besides editing the title, which is not totally clear. I would accept the manuscript in the current format.

We appreciate your positive feedback and valuable comments on our work. We have changed the title to make it clearer as “Regulation of Satiety by Bdnf-e2-expressing Neurons through TrkB Activation in Ventromedial Hypothalamus”.

Reviewer 3 Report

In the current manuscript, Pengcheng Chu and co-workers investigated the role of Bdnf-e2-transcripts expressed in the hypothalamic VMH neurons in regulating energy intake. By employing an array of experiments authors further decipher the mechanism of this regulation by TrkB pathway.  Overall, the study seems interesting with a few modifications that might improve the quality of the paper.

1.  Throughout the manuscript there are various grammatical errors, and wrong indexing of figures example:

line 173: Fig. 1e should be Fig 1f.

line 175: Fig.1c

Line 337: wrong figure number, Extended Data Fig. 6a-c

Line 334, 345, 356:  VPM

Line 372: Extended Data Fig. 9a; there is as such no Fig 9a

Line 506 punctuation error

2.       In the methods section please mention the relevant catalog numbers, source, and viral serotypes used in the current study. This will help others to reproduce the results.

3.       Please clarify what percentage of HFD was used in the current study.

4.       In the introduction, the author mentioned that promoter II transcripts resulted in aggressive behaviors. While given that VMH is known for aggressive behaviors in mice, elevated energy intake in mutant mice might be a secondary effect of BDNF promoter manipulation.

5      Figure 2a: Given that Bdnf-ef-/- is congenitally deleted, then why did food intake and body weight effects only evident after 6 days?

6.       Figure 2: There might be a separate pathway through which the body may regulate BAT vs core body temperature. In this experiment, the author should provide thermal imaging or BAT temperature to confirm their finding. Also, 1h time points of temperature measurement seem very long, a few more time points would be the best options.  Additionally, the author should provide Bdnf-e1 -/- data to ensure the experiment is working.

7.       Figure 5 Please provide the control AAV mCherry data for the CNO experiment, as the paired comparison does not rule out the injection damage and secondary effect at the site of injection damage.

7.       Please discuss the interaction mechanism between Ob/Ob and DIO models with TrkB-ago.

Author Response

Reviewer #3

In the current manuscript, Pengcheng Chu and co-workers investigated the role of Bdnf-e2-transcripts expressed in the hypothalamic VMH neurons in regulating energy intake. By employing an array of experiments authors further decipher the mechanism of this regulation by TrkB pathway.  Overall, the study seems interesting with a few modifications that might improve the quality of the paper.

We thank for the reviewer’s comments as our findings are ‘interesting with a few modifications that might improve the quality of the paper’. We have revised the manuscript based on the suggestions from the reviewer and hope the revision could meet his/her satisfaction.

1. Throughout the manuscript there are various grammatical errors, and wrong indexing of figures example:

line 173: Fig. 1e should be Fig 1f.

line 175: Fig.1c

Line 337: wrong figure number, Extended Data Fig. 6a-c

Line 334, 345, 356:  VPM

Line 372: Extended Data Fig. 9a; there is as such no Fig 9a

Line 506 punctuation error

We appreciate you bringing the errors to our attention. We have reviewed the manuscript and corrected the grammatical errors, punctuation errors, and incorrect figure indexing.

2. In the methods section please mention the relevant catalog numbers, source, and viral serotypes used in the current study. This will help others to reproduce the results.

Thank you for this suggestion. We have now included the catalog numbers, sources, and viral serotypes used in the current study to facilitate reproducibility.

3. Please clarify what percentage of HFD was used in the current study.

We have clarified the “percentage of HFD” used in the current study in both the methods and results sections. We used a 45% fat HFD in the food consumption test and a 60% fat HFD to induce DIO in mice.

4. In the introduction, the author mentioned that promoter II transcripts resulted in aggressive behaviors. While given that VMH is known for aggressive behaviors in mice, elevated energy intake in mutant mice might be a secondary effect of BDNF promoter manipulation.

Thank you for the insightful comment. Previous research has indicated that the ventrolateral subdivision of the ventromedial hypothalamus (VMH) [1], as well as the presence of PR-expressing neurons in the VMH [2], may play a role in male mice exhibiting aggressive behavior. Notably, the ablation of PR-expressing neurons in VMH showed no hyperphagia or increased body weight in mice [2]. Interestingly, SF1-expressing neurons in VMH may influence satiety, but the manipulation of these neurons do not influence anxiety, and no related aggressive behavior has been reported [3]. These findings suggest that regulation of aggressive behavior and satiety may occur through separate pathways. Moreover, there is currently no evidence to suggest that aggressive behavior itself could induce food intake. Thus, it is possible that Bdnf-e2 neurons contribute to both aggressive behavior and energy balance through distinct mechanisms. Further investigation of single cell transcriptome in the VMH coupled with Bdnf-e2 expression may help testing this hypothesis.

5. Figure 2a: Given that Bdnf-e2-/- is congenitally deleted, then why did food intake and body weight effects only evident after 6 days?

Figure 2a shows the changes in body weight and food intake during the initial phase of weight gain in the mice that were tested (from “no difference” the difference appeared, in comparison with the wild type mice). As the reviewer pointed out, the e2 transcript was congenitally deleted, but we did not observe any overweight phenotype in mice before 6 weeks of age. In fact, we only observed the overweight phenotype in mice starting from 8 weeks of age. This may suggest the presence of a developmental regulatory mechanism, which could potentially be an interesting avenue for further investigation.

6. Figure 2: There might be a separate pathway through which the body may regulate BAT vs core body temperature. In this experiment, the author should provide thermal imaging or BAT temperature to confirm their finding. Also, 1h time points of temperature measurement seem very long, a few more time points would be the best options. Additionally, the author should provide Bdnf-e1 -/- data to ensure the experiment is working.

The experiment illustrated in Figure 2 aims to check the thermogenesis ability of Bdnf-e2-/- mice through the functional and molecular test. Rodents would elevate BAT thermogenesis to maintain core (rectal) temperature against cold conditions. Thus, core (rectal) temperature after cold exposure is a critical indicator that reflects thermogenesis ability (see [4] for the reference). However, using temperature probes or thermal imaging to detect brown adipose tissue (BAT) involves both heat production and dissipation factors, which could not fully represent thermogenesis ability. In addition, as one hour cold is a strong challenge for testing thermogenesis ability (see the reference of decreased thermogenesis in Bdnf-e1-/- mice [5]), the absence of differences suggests that there are no issues with thermogenesis ability. To minimize stress to the mice from rectal temperature measurements, we did not (and should not) measure them repeatedly within an hour.

7. Figure 5 Please provide the control AAV mCherry data for the CNO experiment, as the paired comparison does not rule out the injection damage and secondary effect at the site of injection damage.

We have conducted experiments to assess the impact of injection damage of CNO on our results(Extended Data Fig. 7), which showed no difference of food intake between the groups of Bdnf-e2-/- mice with or without application of CNO. Similarly, a previous study found that mice with AAV-control virus infection in VMH did not exhibit abnormal food intake [3]. Moreover, we were concerned about injection damage and therefore conducted experiments starting four weeks after the mice had recovered from surgery. At this time, mice injected with AAV-control virus (e2 in Figure 4) still ate significantly more than WT mice (similar to 5-6g as in other batches of e2 mice). Mice injected with AAV-HM3D also ate 5-6g. Therefore, there should be no injection damage.

8. Please discuss the interaction mechanism between Ob/Ob and DIO models with TrkB-ago.

We found the decrease of food intake and body weight in hyperphagic leptin-deficient (Ob/Ob) mice after TrkB-ago treatment. This finding is consistent with two independent studies suggesting that leptin can regulate energy homeostasis, such as homeostatic feeding, through TrkB signaling in a few of hypothalamic regions [6,7]. Tsao.et al reported that TrkB agonist NT-4 is functional in rescuing obesity phenotype in (DIO) models [8]. Similarly, the treatment with TrkB-ago induced a weight loss in diet-induced obese (DIO, 60% fat HFD) mice. We have added these points in the last section of Results.

Reference

  1. Lin, D.; Boyle, M.P.; Dollar, P.; Lee, H.; Lein, E.S.; Perona, P.; Anderson, D.J. Functional identification of an aggression locus in the mouse hypothalamus. Nature 2011, 470, 221-226, doi:10.1038/nature09736.
  2. Yang, C.F.; Chiang, M.C.; Gray, D.C.; Prabhakaran, M.; Alvarado, M.; Juntti, S.A.; Unger, E.K.; Wells, J.A.; Shah, N.M. Sexually dimorphic neurons in the ventromedial hypothalamus govern mating in both sexes and aggression in males. Cell 2013, 153, 896-909, doi:10.1016/j.cell.2013.04.017.
  3. Zhang, J.; Chen, D.; Sweeney, P.; Yang, Y. An excitatory ventromedial hypothalamus to paraventricular thalamus circuit that suppresses food intake. Nature communications 2020, 11, 6326, doi:10.1038/s41467-020-20093-4.
  4. An, J.J.; Liao, G.Y.; Kinney, C.E.; Sahibzada, N.; Xu, B. Discrete BDNF Neurons in the Paraventricular Hypothalamus Control Feeding and Energy Expenditure. Cell Metab 2015, 22, 175-188, doi:10.1016/j.cmet.2015.05.008.
  5. You, H.; Chu, P.; Guo, W.; Lu, B. A subpopulation of Bdnf-e1–expressing glutamatergic neurons in the lateral hypothalamus critical for thermogenesis control. Molecular Metabolism 2020, 31, 109-123, doi:10.1016/j.molmet.2019.11.013.
  6. Wang, P.; Loh, K.H.; Wu, M.; Morgan, D.A.; Schneeberger, M.; Yu, X.; Chi, J.; Kosse, C.; Kim, D.; Rahmouni, K.; et al. A leptin-BDNF pathway regulating sympathetic innervation of adipose tissue. Nature 2020, 583, 839-844, doi:10.1038/s41586-020-2527-y.
  7. Liao, G.Y.; Kinney, C.E.; An, J.J.; Xu, B. TrkB-expressing neurons in the dorsomedial hypothalamus are necessary and sufficient to suppress homeostatic feeding. Proceedings of the National Academy of Sciences of the United States of America 2019, 116, 3256-3261, doi:10.1073/pnas.1815744116.
  8. Tsao, D.; Thomsen, H.K.; Chou, J.; Stratton, J.; Hagen, M.; Loo, C.; Garcia, C.; Sloane, D.L.; Rosenthal, A.; Lin, J.C. TrkB agonists ameliorate obesity and associated metabolic conditions in mice. Endocrinology 2008, 149, 1038-1048, doi:10.1210/en.2007-1166.

Round 2

Reviewer 3 Report

Thank you for considering the suggestions.